

# Heterologous expression of a Glyoxalase I gene from sugarcane confers tolerance to several environmental stresses in bacteria

Qibin Wu[1], Shiwu Gao[1], Yong-Bao Pan[2], Yachun Su[1], Michael P. Grisham[2], Jinlong Guo[1], Liping Xu[1] and Youxiong Que[1]

[1] Fujian Agriculture and Forestry University, Key Laboratory of Sugarcane Biology and Genetic Breeding, Fuzhou, Fujian, China
[2] USDA-ARS, Sugarcane Research Unit, Houma, LA, USA

## ABSTRACT

Glyoxalase I belongs to the glyoxalase system that detoxifies methylglyoxal (MG), a cytotoxic by-product produced mainly from triose phosphates. The concentration of MG increases rapidly under stress conditions. In this study, a novel glyoxalase I gene, designated as *SoGloI* was identified from sugarcane. *SoGloI* had a size of 1,091 bp with one open reading frame (ORF) of 885 bp encoding a protein of 294 amino acids. SoGloI was predicted as a $Ni^{2+}$-dependent GLOI protein with two typical glyoxalase domains at positions 28–149 and 159–283, respectively. *SoGloI* was cloned into an expression plasmid vector, and the Trx-His-S-tag SoGloI protein produced in *Escherichia coli* was about 51 kDa. The recombinant *E. coli* cells expressing *SoGloI* compared to the control grew faster and tolerated higher concentrations of NaCl, $CuCl_2$, $CdCl_2$, or $ZnSO_4$. *SoGloI* ubiquitously expressed in various sugarcane tissues. The expression was up-regulated under the treatments of NaCl, $CuCl_2$, $CdCl_2$, $ZnSO_4$ and abscisic acid (ABA), or under simulated biotic stress conditions upon exposure to salicylic acid (SA) and methyl jasmonate (MeJA). SoGloI activity steadily increased when sugarcane was subjected to NaCl, $CuCl_2$, $CdCl_2$, or $ZnSO_4$ treatments. Subcellular observations indicated that the SoGloI protein was located in both cytosol and nucleus. These results suggest that the *SoGloI* gene may play an important role in sugarcane's response to various biotic and abiotic stresses.

## INTRODUCTION

Ubiquitously occurring in nature, the glyoxalase pathway involves a two-step catalytic reaction. In the first step, glyoxalase I (GLOI, lactoylglutathione lyase; EC 4.4.1.5) catalyzes the isomerization of hemithioacetal formed spontaneously between methylglyoxal (MG) and reduces glutathione (GSH) to S-D-lactoylglutathione (S-LG) (*Thornalley, 1993*). In the second step, S-LG is hydrolyzed by glyoxalase II (GLOII, hydroxyacylglutathione hydrolase; EC 3.1.2.6) to produce GSH and D-lactate (*Yadav et al., 2005a*; *Yadav et al., 2005b*). Under normal physiological conditions, MG is produced primarily through

Corresponding authors
Liping Xu, xlpmail@126.com
Youxiong Que, queyouxiong@126.com

glycolysis at the triose-phosphate step (*Phillips & Thornalley, 1993*), and to a much lesser extent, through catabolism of amino acids (threonine and glycine) and acetone (*Yadav et al., 2005a*; *Yadav et al., 2005b*; *Yadav et al., 2007*). Under abiotic stresses, however, the concentration of MG in plants can significantly increase by 2–6 folds (*Yadav et al., 2005a*). The high-level accumulation of MG is toxic to cells, as it can react with DNA to form modified guanylate residues (*Papoulis, Al-Abed & Bucala, 1995*). MG also can react with proteins to form glycosylamine derivatives of arginine, lysine and hemithioacetal with cysteine residues (*Lo et al., 1994*). Apart from the direct effect of MG, its intermediate compound S-LG, a substrate for glyoxalase II, is also cytotoxic at higher concentrations by inhibiting DNA synthesis (*Thornalley, 1996*). Therefore, glutathione-based detoxification of harmful metabolites is one of the main roles of both glyoxalase enzymes (*Thornalley, 1990*).

The glyoxalase I enzyme is broadly categorized into $Zn^{2+}$-or $Ni^{2+}$-dependent class of metal activation. Previous studies have been showed that the $Zn^{2+}$-dependent GLOI enzymes are thought to be of eukaryotic origin (*Frickel et al., 2001*; *Ridderström & Mannervik, 1996*), while $Ni^{2+}$-dependent GLOI enzymes are thought to be of prokaryotic origin (*Sukdeo et al., 2004*). The coexistence of both $Zn^{2+}$-dependent and $Ni^{2+}$-dependent GLOI enzymes in *Pseudomonas aeruginosa* (*Sukdeo & Honek, 2007*) and the characterization of a $Ni^{2+}$-dependent GLOI enzyme from rice (*Mustafiz et al., 2014*) have led to the discouragement of the view that $Zn^{2+}$-dependent GLOI belongs to eukaryotes and $Ni^{2+}$-dependent GLOI exists only in prokaryotes (*Jain et al., 2016*). In plants, the metal specificity of each member of the GLOI family is an important determinant of its catalytic efficiency (*Kaur et al., 2017*; *Mustafiz et al., 2014*).

The glyoxalase system has been widely studied in animals and microbes (*Price & Knight, 2009*; *Thornalley, 2003*). Some studies have suggested that the glyoxalase system is associated with clinical disorders, such as cancer and diabetes (*Price & Knight, 2009*; *Thornalley, 2003*). To date, *GLOI* has been cloned and characterized from several plant species, including *Oryza sativa* (*Kaur et al., 2017*; *Mustafiz et al., 2014*; *Usui et al., 2001*; *Zeng et al., 2016*), *Glycine max* (*Skipsey et al., 2000*), *Zea mays* (*Chen et al., 2004*), *Triticum aestivum* (*Lin et al., 2010*), *Triticum medicata* (*Johansen, Svendsen & Rasmussen, 2000*), *Brassica juncea* (*Veena, Reddy & Sopory, 1999*), *Solanum lycopersicum* (*Espartero, Sanchez-Aguayo & Pardo, 1995*), *Allium cepa* (*Hossain & Fujita, 2009*), and *Beta vulgaris* M14 (*Wu et al., 2013a*). Studies have shown that GLOI activity is up-regulated in response to light and phytohormones (*Chakravarty & Sopory, 1988*; *Deswal & Sopory, 1999*; *Sethi, Basu & Guha-Mukherjee, 1988*). Moreover, enzyme activity and transcription of GLOI are up-regulated in response to abiotic and biotic stresses, such as NaCl, heavy metals, mannitol, MG, abscisic acid or pathogen attack (*Espartero, Sanchez-Aguayo & Pardo, 1995*; *Ghosh & Islam, 2016*; *Jain et al., 2016*; *Kaur et al., 2017*; *Lin et al., 2010*; *Mustafiz et al., 2011*; *Mustafiz et al., 2014*; *Singla-Pareek, Reddy & Sopory, 2003*; *Singla-Pareek et al., 2006*; *Veena, Reddy & Sopory, 1999*). Other *GLOI* genes such as *Gly I*, *GLX1*, B*v* M14-*glyoxalase I*, *TaGly I*, *OsGLYI*-11.2, *OsGly I*, and *OsGlyI*-8 isolated from *B. juncea* (*Veena, Reddy & Sopory, 1999*), *S. lycopersicum* (*Espartero, Sanchez-Aguayo & Pardo, 1995*), *B. vulgaris* M14 (*Wu et al., 2013a*), *T. aestivum* (*Lin et al., 2010*), and rice (*Kaur et al., 2017*; *Mustafiz et al., 2014*;

*Zeng et al., 2016*) respectively, have been associated with significant tolerance to salt stress. *Singla-Pareek et al. (2006)* have shown that *Gly I* from *B. Juncea* enhances the tolerance to heavy metals, such as zinc, cadmium or lead. *Lin et al. (2010)* also have shown that *TaGly I* was up-regulated under $ZnCl_2$ treatment. *Mustafiz et al. (2011)* reported a high level of stress inducibility of $Ni^{2+}$-dependent *GlxI* encoding *AtGLYI3* and *AtGLYI6* in response to salt, drought, wounding, cold and heat treatments in both *Arabidopsis thaliana* roots and shoots, whereas $Zn^{2+}$-dependent *GlxI* encoding *AtGLYI2* was found to be heat inducible. Transgenic tobacco plants over-expressing *GlyI* were highly tolerant to MG, salt, or zinc and were able to grow, flower, and produce viable seeds (*Singla-Pareek, Reddy & Sopory, 2003*; *Singla-Pareek et al., 2006*; *Veena, Reddy & Sopory, 1999*). *Zeng et al. (2016)* reported that *OsGly I* was markedly up-regulated in response to NaCl, $ZnCl_2$ and mannitol in rice seedlings. Compared to the wild-type plants, the *OsGly I*-overexpression transgenic rice lines showed increased glyoxalase enzyme activity, decreased MG level, improved tolerance to NaCl, $ZnCl_2$ and mannitol, and produced higher rates of seed setting and higher yields. Another $Ni^{2+}$-dependent *GlxI* gene, *OsGLYI*-11.2, was found to be highly stress inducible amongst the rice glyoxalase family (*Mustafiz et al., 2014*). Similarly, $Ni^{2+}$-dependent *GlxI* genes from *Arabidopsis* were found to be more responsive to various stress treatments compared to $Zn^{2+}$-dependent *GlxI* genes (*Mustafiz et al., 2011*). All these findings suggest that the *GLOI* genes may play an important role in response to stresses.

Sugarcane (*Saccharum* spp. hybrids) is a cash crop mainly used for sugar, biofuel and other food industries such as industrial alcohol in tropical and subtropical regions. It is one of the world's largest crops. According to the FAO, sugarcane was cultivated in 101 countries on about 26.1 million hectares of land in 2012 (*Que et al., 2014*). However, the yields of sugarcane are often influenced by many diseases and various environmental stresses, such as smut, rust, ratoon stunting disease (RSD), salt, heavy metal and drought. Sugarcane is reportedly susceptible to salt and shows toxicity symptoms, low sprout emergence, nutritional imbalance, and overall biomass reduction (*Akhtar, Wahid & Rasul, 2003*; *Plaut, Meinzer & Federman, 2000*; *Wahid, Rao & Rasul, 1997*). Though sugarcane plants can overcome a short period of water deficit during the late, sucrose accumulating growth stage, an extended period of drought can cause a significant loss in cane and sugar yields (*Begcy et al., 2012*). RSD causes significant yield losses, 12%–37% under normal conditions and up to 60% under drought conditions. Moreover, it may also lead to variety deterioration (*Bailey & Bechet, 1997*; *James, 1996*; *Que et al., 2008*). Sugarcane smut also causes serious losses in cane and sugar yields (*Hoy et al., 1986*; *Padmanaban, Alexander & Shanmugan, 1998*; *Que et al., 2012*).

In our previous study, we constructed a sugarcane cDNA library from *Sporisorium scitamineum*-infected buds (*Wu et al., 2013b*). An expressed sequence tag of 613 bp (GenBank Accession Number: CA140600.1) had a high similarity to the *GLOI* gene of *Zea mays* (GenBank Accession Number: EU966885.1) (*Wu et al., 2013b*). To study stress response of *GLOI* in sugarcane, we cloned the entire sugarcane Glyoxalase I gene, designated as *SoGloI*. We determined the sub-cellular location of the *SoGloI*'s protein using tobacco protoplasts and investigated growth patterns of *Escherichia coli* Rosetta cells producing the *SoGloI* recombinant protein in response to salt and heavy metal ion stresses.

**Table 1  Primers used in the present study.**

| Primer | Sequence (5′–3′) | Application |
|---|---|---|
| G-F | AGCCAGAAGAAAGGGAGC | RT-PCR |
| G-R | GTTTCCGCCTTGATGAAC | |
| G-32aF | CGGAATTCATGGCAACTGGTAGTGAAG | Prokaryotic expression vector construction |
| G-32aR | CCCTCGAGTCAGTGAAGTTCCCTGAG | |
| G-QF | TGGACCGCACAATCAAATACTACAC | |
| G-QR | GTATGACATTGGAACGGGCTTTG | |
| 25S-QF | GCAGCCAAGCGTTCATAGC | RT-qPCR |
| 25S-QR | CCTATTGGTGGGTGAACAATCC | |
| GAPDH-QF | CACGGCCACTGGAAGCA | |
| GAPDH-QR | TCCTCAGGGTTCCTGATGCC | |
| PCX-GFP-F | ATGGTGAGCAAGGGCGAGGAG | |
| GLO-LAP1-R | ACTACCAGTTGCCATCTTGTACAGC | Fusion PCR primers for sub-cellular localization |
| GLO-LAP2-F | GCTGTACAAGATGGCAACTGGTAGT | |
| GLO-R | TCAGTGAAGTTCCCTGAGGAAGTCG | |

We also assessed *SoGloI* expression and glyoxalase I enzyme in sugarcane in response to simulated biotic and abiotic stresses. The results provided valuable information for the improvement of stress resistance in sugarcane.

# MATERIALS & METHODS

## Plant material

Sugarcane genotype YCE 05-179 was used in this study. Plants were maintained in a genetic nursery at the Key Laboratory of Sugarcane Biology and Genetic Breeding, Ministry of Agriculture, Fujian Agriculture and Forestry University, Fuzhou, China. In addition, tissue culture-derived young, healthy plantlets of YCE 05-179 were also involved in the study.

## Cloning of *SoGloI* gene

A BLAST search of the sugarcane database (taxid: 286192; taxid: 4547; taxid: 128810; taxid: 62335) of GenBank (http://www.ncbi.nlm.nih.gov/) was conducted with the sugarcane EST, CA140600.1 (*Wu et al., 2013b*). Several highly homologous sugarcane ESTs were obtained and aligned. A 1,446 bp cDNA of sugarcane *Glo I* gene, designated as *SoGloI*, was assembled using the CAP3 Sequence Assembly Program (http://doua.prabi.fr/software/cap3). *SoGloI* cDNA sequence was amplified in a 25 μL reaction mixture on ABI Veriti96 PCR with primers G-F and G-R (Table 1) under a thermal cycling program: 94 °C, 4 min; 35 cycles of (94 °C, 1 min; 54 °C, 1 min; 72 °C, 1.5 min); and 72 °C, 10 min. Reaction mixture contained 2.5 μL 10× PCR buffer (plus $Mg^{2+}$), 2.5 μL dNTPs (2.5 mM), 1.0 μL first-strand cDNA, 1.0 μL each of forward and reverse primers (10 μM) and 0.125 μL *Ex-Taq* enzyme (5 U μL$^{-1}$) (Takara, Shanghai, China). Amplified *SoGloI* cDNA product was separated through 1% agarose gel electrophoresis, gel purified using Omega Gel Extraction Kit (Omega, Shanghai, China), and cloned into the PMD 18-T vector (Takara, Shanghai, China). Putative recombinants were confirmed by PCR, of which six were sequenced (Shenggong Co., Ltd., Shanghai, China).

## Bioinformatics analysis

The ORF of the full-length *SoGloI* was predicted using the ORF Finder program (http://www.ncbi.nlm.nih.gov/gorf/gorf.html). Molecular weight and isoelectric point was calculated using Compute pI/Mw tool (http://www.expasy.ch/tools/pi_tool.html). Protein domain was predicted using the SMART Program (http://smart.embl-heidelberg.de/). Protein hydrophilicity was analyzed by Protscale Program (http://www.expasy.ch/tools/protscale.html). Prediction of signal peptides and analysis of trans-membrane helix domain were conducted using the SignalP and TMHMM-2.0 Programs (http://www.cbs.dtu.dk/services/). A multiple protein-sequence alignment was carried out using the DNAMAN® Program (*Guo et al., 2012*; *Liu et al., 2010*; *Su et al., 2013*).

## Expression of *SoGloI* in field-grown sugarcane plants

Five tissue samples, white young roots, leaf (+1), leaf sheath (+1), buds (6th-8th from the base), and internodes (6th and 7th from the base) were collected from 7- to 8-month-old plants in the field nursery. All samples except buds were cut into small pieces, wrapped up within tinfoil, and immediately flash-frozen in liquid nitrogen. The frozen samples were stored in a −80 °C freezer until RNA extraction. Total RNA samples were isolated from the frozen buds using the TRIzol® kit (Invitrogen, Carlsbad, CA, USA) according to the manufacturer's instructions. RNA samples were dissolved in diethylpyrocarbonate-treated $H_2O$. RNA concentration was quantified by measuring absorbance at $OD_{260}$ and $OD_{280}$ using Synergy H1 Microplate Reader Multi-Mode (Bio-Tek, Winooski, VT, USA). RNA quality was also assessed by 1.0% denaturing agarose gel electrophoreses.

First-strand cDNA was synthesized from 1 µg RNA in 20 µL reaction mixture by reverse transcription PCR (RT-PCR) using the Prime-Script® 1st Strand cDNA Synthesis Kit (Takara, Shanghai, China) on ABI Veriti96 PCR (ABI, Foster City, CA, USA). For real-time quantitative PCR (RT-qPCR), a total of 1 µg RNA in 20 µL reaction mixtures was used for the first-strand cDNA synthesis using the Prime-Script® RT Reagent Perfect Real Time Kit (Takara, China) on ABI Veriti96 PCR. Phosphoglyceraldehyde dehydrogenase (*GAPDH*) (CA254672) gene was used as the internal control. The primers of GAPDH-QF/GAPDH-QR were listed in Table 1 (*Iskandar et al., 2004*; *Que et al., 2009a*; *Que et al., 2009b*).

## *SoGloI* expression in greenhouse-grown sugarcane plantlets under different stress treatments

In the greenhouse, tissue culture-derived 4-month old healthy plantlets were rinsed with agar medium and transplanted into every other column of 46 mL-wells in two 96-well (8 × 12) plastic trays, containing only distilled water. The water was changed every morning. After 10 days, the plantlets were transplanted to 100-mL flat-bottomed glass tubes and subjected to seven different stress treatments in three replicates. Three plantlets of each treatment was treated with the following solutions: 100 µM methyl jasmonate (MeJA, dissolved with 0.1% (v/v) ethanol and 0.05% (v/v) Tween-20); 5.0 mM salicylic acid (SA, dissolved with 0.05% (v/v) Tween-20); 100 µM abscisic acid (ABA) for 6, 12, and 24 h (*Li et al., 2009*; *Su et al., 2013*); 250 mM NaCl; 500 µM $CdCl_2$; 100 µM $CuCl_2$; or 100 µM $ZnSO_4$ for 12, 24 and 48 h (*Damaj et al., 2010*; *Guo et al., 2012*; *Que et al., 2009a*; *Que et al., 2009b*), respectively. Plantlets grown in distilled water were used as the control.

*SoGloI* expression in response to MeJA, SA, ABA (0, 6, 12, 24 hour-post-treatment or hpt) and NaCl, CdCl$_2$, CuCl$_2$, ZnSO$_4$ (0, 12, 24, 48 hpt) was analyzed by RT-qPCR on a 7500 RT-qPCR system (ABI, Foster City, CA, USA). *25S* rRNA (BQ536525) gene was chosen as internal control and. The primers of 25S-QF/25S-QR were listed in Table 1. The *SoGloI*-specific primer pair of G-QF/G-QR was designed by Primer Premier 5.0 software (Premier Biosoft International, CA) (Table 1). RT-qPCR was carried out with FastStart Universal SYBR Green Master (ROX) (Roche, Shanghai, China) in a 25 µL volume containing 12.5 µL FastStart Universal SYBR Green PCR Master (ROX), 0.5 µM of each primer and 1.0 µL template (100× diluted cDNA). RT-qPCR with sterile dH$_2$O as template was the control. The RT-qPCR thermal cycle program included 2 min at 50 °C; 10 min at 95 °C; and 40 cycles of (15 s at 94 °C and 1 min at 60 °C). The reactions were repeated three times for each sample. The 2$^{-ddCt}$ method was used to calculate relative gene expression levels (*Livak & Schmittgen, 2001*).

## Expression of pET32a-*SoGloI* in *E. coli* Rosetta Cells (DE3)

A *SoGloI* gene fragment was amplified from the cDNA clone with primers G-32aF/G-32aR (Table 1). The *SoGloI* PCR fragment was digested with *EcoR* I and *Xho* I (NEB, USA) and subsequently sub-cloned into the *EcoR* I–*Xho* I sites of pET32a (+) (*Guo et al., 2012*) to produce pET32a-*SoGloI*. After sequence verification, the pET32a-*SoGloI* was transformed into Rosetta Cells (DE3) (Tiangen BioTech Co. Ltd., Sichuan, China). Expression of pET32a-*SoGloI* was induced in 1 mM isopropyl β-D-thiogalactoside (IPTG) for 8 h at 37 °C and analyzed by sodium dodecyl sulfate-polyacrylamide gel electrophoresis (SDS-PAGE) at 0, 2, 4, and 8 h. Non-transformed (blank) and the vector pET32a-transformed Rosetta cells (control) were used as the controls.

Spot assays were performed to assess the response of pET32a-*SoGloI* transformed *E. coli* cells to NaCl, CdCl$_2$, CuCl$_2$ or ZnSO$_4$ treatments. When the *E. coli* culture mixture reached OD$_{600}$ = 0.6, 1 mM IPTG was added into the LB medium and the culture mixture was incubated for 12 h at 28 °C. Then the cultures were first diluted to 0.6 (OD$_{600}$), and further diluted to two levels ($10^{-3}$ and $10^{-4}$) (*Guo et al., 2012*). Thereafter, 10 µL each of the diluted cultures was spotted on LB plates containing 170 µg mL$^{-1}$ chloramphenicol and 80 µg mL$^{-1}$ ampicillin, along with each test chemical. The concentrations of the chemicals used were NaCl at 250, 500 and 750 mM, CdCl$_2$ at 250, 500 and 750 µM, CuCl$_2$ at 250, 500 and 750 µM, and ZnSO$_4$ at 250, 500 and 750 µM, respectively (*Guo et al., 2012*; *Su et al., 2013*). All plates were incubated overnight at 37 °C.

## Assay of sugarcane glyoxalase I enzyme activity

Entire flash-frozen 4-month old plantlets (100 mg wet weight) were pulverized in liquid N$_2$ in a mortar. Protein was extracted with an extraction buffer containing 0.1 M potassium phosphate buffer (PPB, pH 7.5), 50% (v/v) glycerol, 16 mM MgSO$_4$, 0.2 mM Phenylmethanesulfonyl fluoride (PMSF) and 0.2% (v/v) polyvinylpyrrolidone (PVP40). The extract was centrifuged twice at 13,000 rpm at 4 °C for 30 min to obtain the crude protein extract from the supernatant (*Zeng et al., 2016*). The supernatant was used as the cytosolic extract for the assessment of glyoxalase activity, and protein concentration was

determined by the Bradford method (*Bradford, 1976*) using bovine serum albumin (BSA) as the standard. SoGloI activity assay was carried out following Hossain et al. (2009) and *Hasanuzzaman, Hossain & Fujita (2011)*. Briefly, the assay mixture contained 100 mM K-phosphate buffer (PPB, pH 7.0), 15 mM magnesium sulfate, 1.7 mM GSH, and 3.5 mM MG in a final volume of 0.8 mL. Thioester formation was measured by the increase in absorbance at 240 nm for 1 min. The enzyme activity was calculated using an extinction coefficient ($\varepsilon$) of 3.37 mM$^{-1}$ cm$^{-1}$.

## Sub-cellular localization

The *SoGloI* gene was sub-cloned into the *Xcm* I/*Bam*H I restriction sites of pCXSN to construct a fusion protein expression vector of 35S::*SoGloI*::GFP. The *GFP*-containing pCXSN vector was a gift of Songbiao Chen, Institute of Biotechnology, Fujian Academy of Agricultural Sciences. The pCXSN-*SoGloI* recombinant plasmids were transformed into *Agrobacterium tumefaciens* cells, strain GV 3101 (*Chen et al., 2006*). The transgenic GV 3101 cells were inoculated into LB medium containing kanamycin (50 $\mu$g mL$^{-1}$) and rifampicin (34 $\mu$g mL$^{-1}$). The culture was incubated overnight at 28 °C with shaking at 200 rpm. The culture was then centrifuged at 5,000× g to harvest the *Agrobacterium* cells followed by, re-suspension in 10 mM MgCl$_2$ and 10 mM fatty acid methyl ester sulfonate (MES). The concentration of the bacterial suspension was measured and adjusted to OD$_{600}$ = 0.6 with Murashige and Skoog (MS) liquid medium supplemented with 200 mM acetosyringone. The resulting culture was incubated at 28 °C for 3 h (*Yang et al., 2014*). Then, 1 mL of the bacterial culture was infiltrated into 4-week old tobacco leaves with disposable syringes. The injection sites were marked. Injected plants were incubated under a 12 h-light/12 h-dark cycle at 28 °C for three days (*Su et al., 2013*). Then, the protoplasts were isolated from well-expanded leaves following the rice protoplast isolation protocol of *Chen et al. (2006)*. Briefly, the leaves were cut into 1-mm strips and placed in a dish containing 12 mL of K3 medium (3 mM MES, 7 mM CaCl$_2$, 0.35 M mannitol, 0.7 mM NaH$_2$PO$_4$, 0.35 M sorbitol, 20 mM KCl, pH 5.6) supplemented with 0.4 M sucrose, 1.5% cellulase R-10 (Yakult Honsha, Japan) and 0.3% macerozyme R-10 (Yakult). The leaf tissue was vacuum-infiltrated for 30 min at 20 mm Hg and digested at room temperature with gentle shaking for 4 h to produce protoplasts. Then, the K3 medium was replaced with 12 mL of W5 solution (2 mM MES, 154 mM NaCl, 125 mM CaCl2, 5 mM KCl, pH 5.8). The protoplasts were collected by centrifugation at 300× g for 4 min at 4 °C and re-suspended in 1 mL WI solution (4 mM MES, 0.5 M mannitol, 20 mM KCl, pH 5.7). The sub-cellular location of the *SoGloI* gene was observed using fluorescence microscopy (Ci-L; Nikon, Tokyo, Japan).

## RESULTS

### Sequence analysis of *SoGloI*

The full-length of *SoGloI* (GenBank Accession Number: KC857628) was 1,091 bp with one ORF of 885 bp (Fig. S1). The deduced SoGloI protein had 294 amino acids with a predicted molecular mass of 32.9 kDa and a pI value of 5.45. Two glyoxalase domains were found, located at 28–149 and 159–283, respectively (Fig. S2). The ProtScale predicted that SoGloI
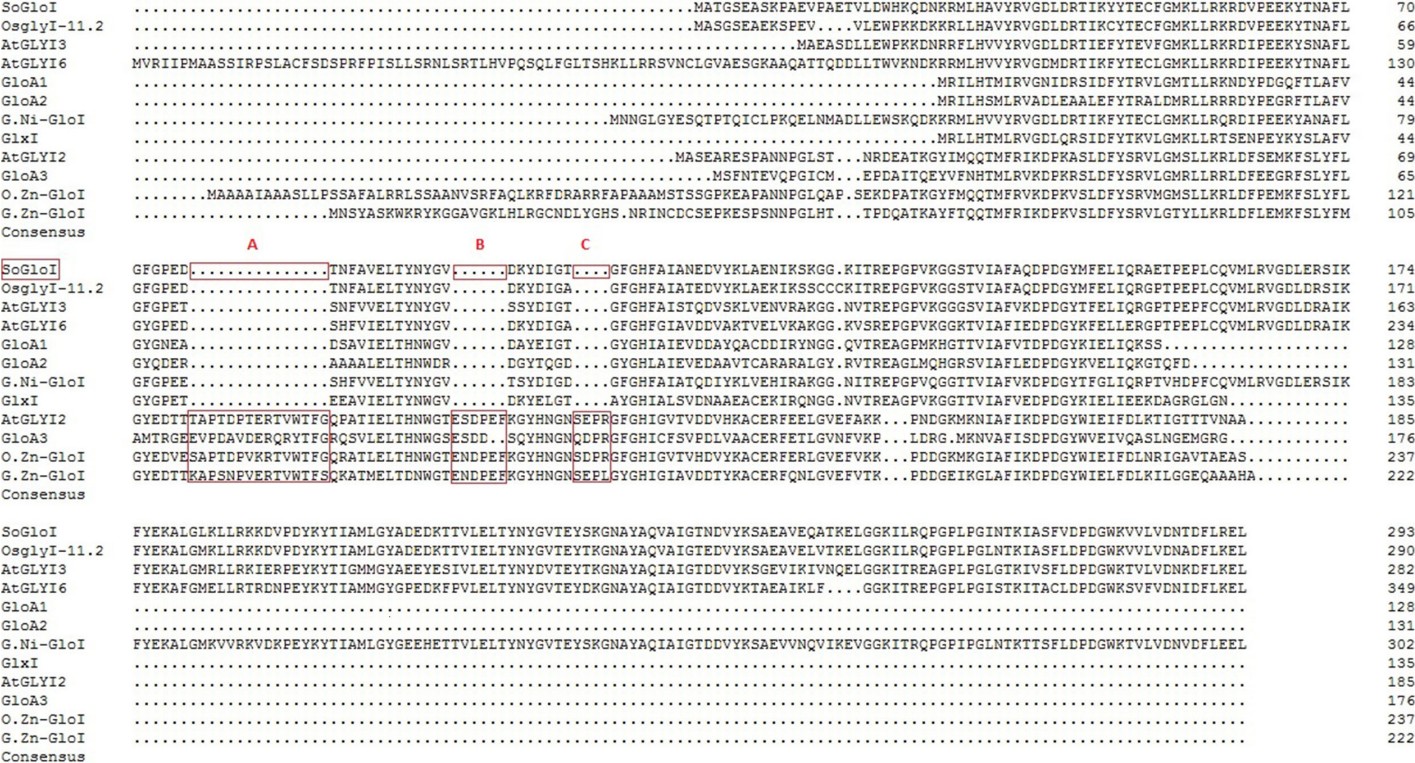

**Figure 1  Multiple protein sequence alignment of SoGloI protein with glyoxalase I proteins from other species.** The species name and accession numbers are as follows: *Oryza sativa* (OsglyI-11.2 and O. Zn-GloI with Acc. No. Os08g09250 and EEC78918.1), *Arabidopsis thaliana* (AtGLYI2, AtGLYI3 and AtGLYI6 with Acc. No. NP_172291.1, NP_172648.1, and NP_176896.1 respectively), *Pseudomonas aeruginosa* (GloA1, GloA2 and GloA3 with Acc. No. AAG06912, AAG04099 and AAG08496 respectively), *Glycine max* (G. Zn-GloI and G. Ni-GloI with Acc. No. XP_003539194.1 and XP_003528689.1) and *Escherichia coli* (GlxI with Acc. No. AAC27133). The letters A, B and C represent extended sequences found in $Zn^{2+}$-dependent GLOI, but absent in $Ni^{2+}$-dependent GLOI.

was a hydrophilic protein (Fig. S3). TMHMM Server v.2.0 and SignalP 4.1 Server program predicted that SoGloI was not a trans-membrane protein (Figs. S4, S5).

In order to predict the metal dependency of SoGloI protein, a multiple sequence alignment of SoGloI with other known GloI proteins from *Oryza sativa* (OsglyI-11.2 and O. Zn-GloI), *Arabidopsis thaliana* (AtGLYI2, AtGLYI3 and AtGLYI6), *Pseudomonas aeruginosa* (GloA1, GloA2 and GloA3), *Glycine max* (G. Zn-GloI and G. Ni-GloI) and *Escherichia coli* (GlxI) was done using DNAMAN® (Fig. 1). SoGloI shared 87.8%, 69.7%, 57.3%, and 72.0% amino acid sequence identities with OsglyI-11.2, AtGLYI3, AtGLYI6, and G. Ni-GloI, respectively (Fig. 1). Extended amino acid sequences (the letters A, B and C in Fig. 1) of OsglyI-11.2, AtGLYI3, AtGLYI6, GloA1, GloA2, G. Ni-GloI, and GlxI were missing in SoGloI, a characteristic of $Ni^{2+}$-dependent GloI. Contrarily, these extended amino acid sequences were present in AtGLYI2, GloA3, O. Zn-GloI and G. Zn-GloI, which were $Zn^{2+}$-dependent GloI.

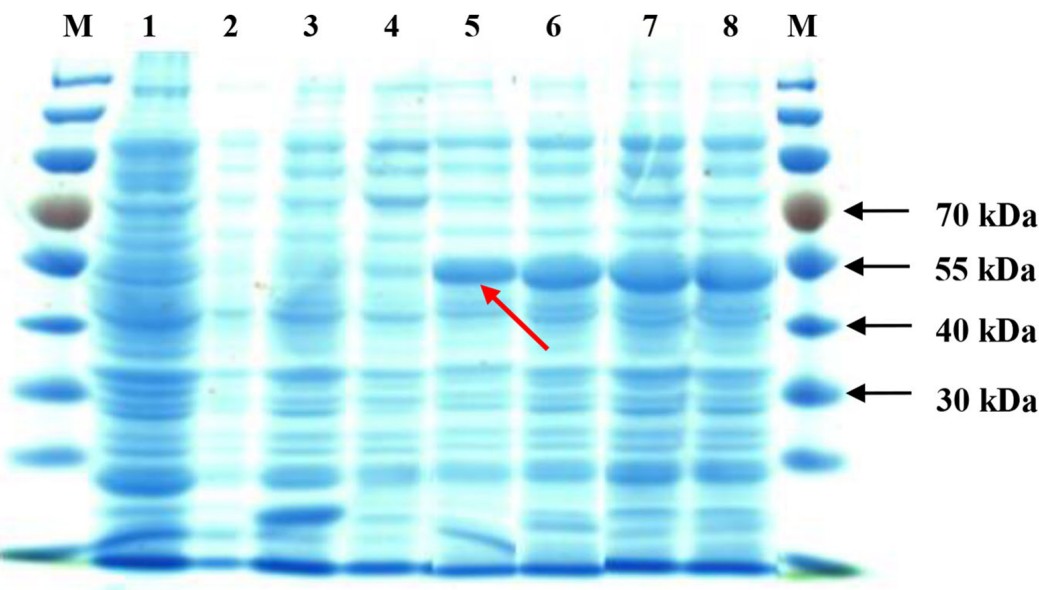

**Figure 2** **Heterologous expression of recombinant SoGloI in *Escherichia coli* Rosetta cells (red arrow).** Lane designation: M, protein size makers with 30, 40, 55, and 70 kDa indicated; 1, blank; 2, control without IPTG induction; 3, control with induction for 8 h; 4, pET 32a-*SoGloI* without induction; 5 to 8, pET32a-*SoGloI* with induction for 2, 4, 6 and 8 h, respectively.

## Expression of *SoGloI* in *E. coli*

Upon IPTG induction, the recombinant *SoGloI* gene expressed well in Rosetta cells (Fig. 2, Lanes 5–8) to yield SoGloI protein that was 51 kDa in size and carried a Trx-His-S-tag of 18.3 kDa. Moreover, gradually increased amounts of SoGloI protein were also observed when the IPTG induction was extended from 2 h to 8 h.

In the spot assays, *SoGloI*-expressing Rosetta cells (Figs. 3A–3E, Row a) grew more rapidly on LB agar plates containing NaCl, $CuCl_2$, $CdCl_2$ or $ZnSO_4$ than control Rosetta cells (pET 32a) (Figs. 3A–3E, Row b). Compared to control cells, *SoGloI*-expressing cells tolerated salt up to 250 mM (Fig. 3B) as well as heavy metal ions up to 750 μM (Figs. 3C–3E). Consistently, *SoGloI*-expressing cells also grew faster than the control cells in LB liquid media containing 250 mM NaCl, 750 μM $CuCl_2$, 750 μM $CdCl_2$ or 750 μM $ZnSO_4$ (Fig. 4). These results may demonstrate that the recombinant SoGloI protein enhanced the growth of Rosetta cells under stress conditions.

## Expression patterns of *SoGloI* in sugarcane tissues

RT-qPCR was conducted to detect both tissue-specific and stress-related expression of *SoGloI*. The *SoGloI* gene was ubiquitously expressed in five tissues of 7- to 8-month old plants collected from the field. The highest level was detected in buds, followed by leaves, roots, leaf sheaths, and internodes (Fig. 5A).

*SoGloI* expression patterns in healthy 4-month old plantlets under NaCl, $CuCl_2$, $CdCl_2$, $ZnSO_4$, SA, MeJA, and ABA treatments were shown in Figs. 5B and 5C. Under NaCl, $CuCl_2$, $CdCl_2$, and $ZnSO_4$ treatments, *SoGloI* expression was up-regulated steadily from 0

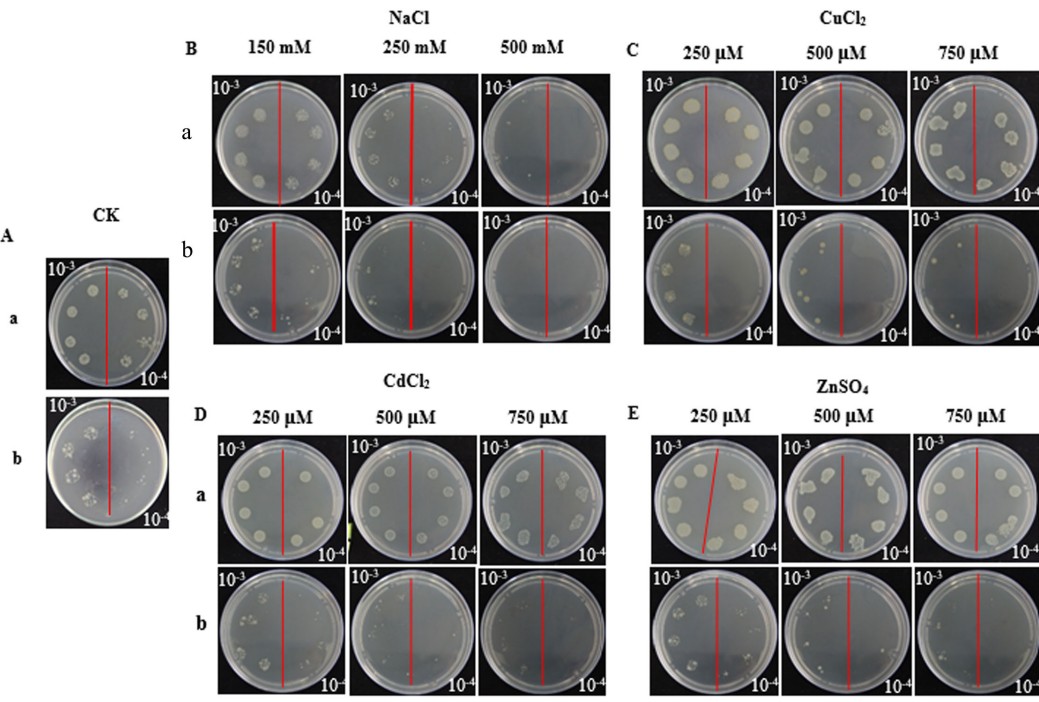

**Figure 3 Growth of pET32a- *SoGloI* -expressing (row a) and pET 32a (row b) *Escherichia coli* Rosetta cells on LB agar plates with various concentrations of salt or heavy metal ions indicated above the figures.** (A) Control; (B) NaCl; (C) CuCl$_2$; (D) CdCl$_2$, and (E) ZnSO$_4$.

to 48 hpt. The peak level of *SoGloI* expression was about 3.1-, 2.9- 2.8- and 1.9-fold of the level in control, respectively (Fig. 5B). In contrast, under SA and MeJA treatments, *SoGloI* expression decreased after peaking at 6 hpt. The maximum level of *SoGloI* expression was detected at 6 hpt, which was about 2.6- and 2.1-fold of the level in control, respectively (Fig. 5C). Similarly, the peak level of *SoGloI* expression was detected at 12 hpt under ABA treatments, which was 2.4-fold of the level in control (Fig. 5C). Thus, *SoGloI* gene has been found to provide tolerance to multiple abiotic stresses.

## Glyoxalase I activity in sugarcane under NaCl, CuCl$_2$, CdCl$_2$ or ZnSO$_4$ treatment

As shown in Fig. 5D, under NaCl, CuCl$_2$, CdCl$_2$, and ZnSO$_4$ treatments, the glyoxalase I activity was increased steadily from 0 to 48 hpt. Under a 250 mM NaCl treatment, the glyoxalase I activity was about 1.8-, 2.2-, and 2.3-fold at 12, 24, and 48 hpt comparing to control, respectively. At 48 hpt, the level of glyoxalase I activity reached 0.3230 $\mu$mol min$^{-1}$ mg$^{-1}$. Under a 750 $\mu$M CuCl$_2$ treatment, the level of glyoxalase I activity was about 2.0-, 2.7-, and 3.0-fold of the level in control, with 0.4128 $\mu$mol min$^{-1}$ mg$^{-1}$ protein produced at 48 hpt. Similarly, under a 750 $\mu$M CdCl$_2$ treatment, the glyoxalase I activity was 2.1-, 3.1- and 4.2-fold comparing to control, with a highest activity of 0.5730 $\mu$mol min$^{-1}$ mg$^{-1}$ protein at 48 hpt. Under 750 $\mu$M ZnSO$_4$ treatment, the glyoxalase I activity at 12, 24, and 48 hpt was about 1.3-, 2.6-, and 3.1-fold of the level at 0 hpt, and 0.2883 $\mu$mol min$^{-1}$ mg$^{-1}$

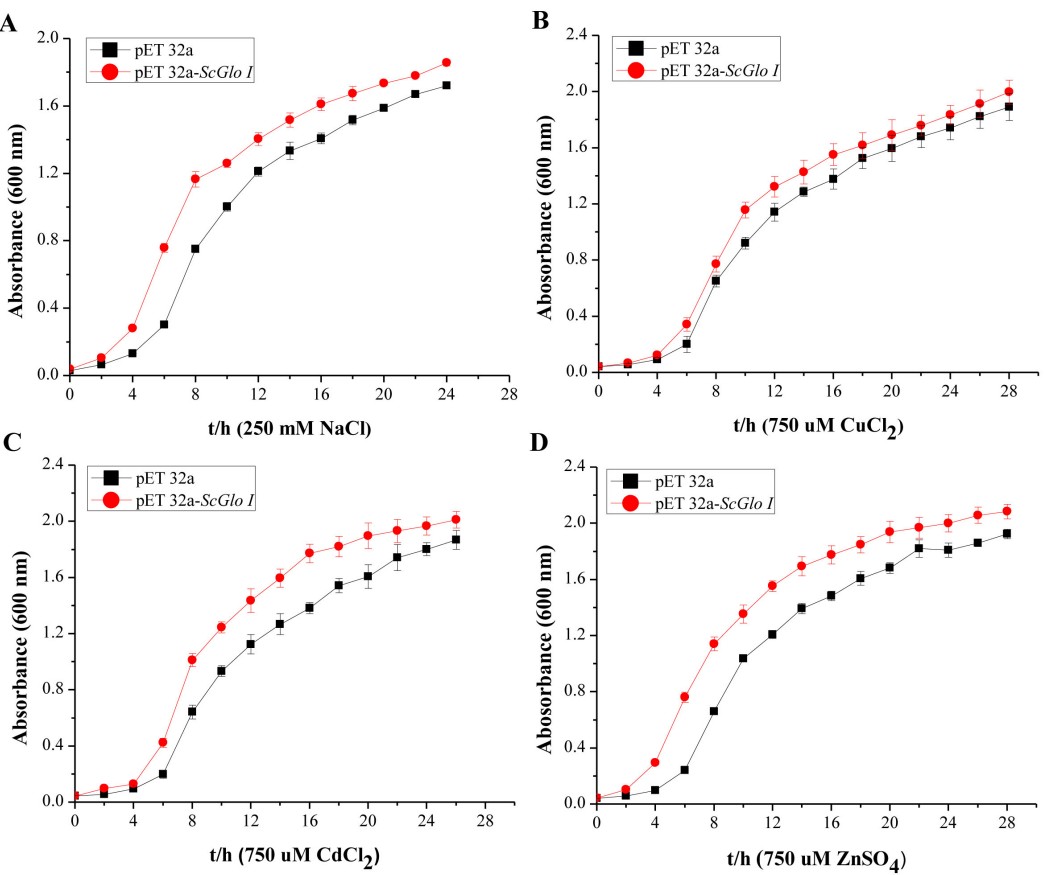

**Figure 4 Growth assessments of SoGloI overexpressing *Escherichia coli* Rosetta cells under salt and heavy metal treatments.** (A) under 250 mM NaCl treatment from 0 h to 24 h; (B) under 750 μM CuCl$_2$ treatment from 0 h to 28 h; (C) under 750 μM CdCl$_2$ treatment from 0 h to 26 h; (D) under 750 μM ZnSO$_4$ treatment from 0 h to 28 h. All data were measured every two hours. All data points are mean ± SE ($n = 3$).

protein produced at 48 hpt. Thus, glyoxalase I activity was increased in varying degrees under salt and heavy metal ions stress conditions.

## Determination of subcellular localization of ScGloI

To further understand the function of *SoGloI* gene, its subcellular localization was determined. The *SoGloI* gene was inserted into a plant expression vector *pCXSN* between the 35*S* promoter and *GFP*. The recombinant pCXSN-*SoGloI*-GFP construct was then introduced into tobacco leaves through *Agrobacterium*-mediated transformation. As shown in Fig. 6, green fluorescence signals were observable in the cytosol and nucleus of both pCXSN-*SoGloI*-GFP and the pCXSN-GFP transformed tobacco protoplasts.

## DISCUSSION

Glyoxalase I functions to detoxify the potent cytotoxic compound MG (*Thornalley, 1993*). In response to stress conditions, cells undergo active metabolism to produce more MG

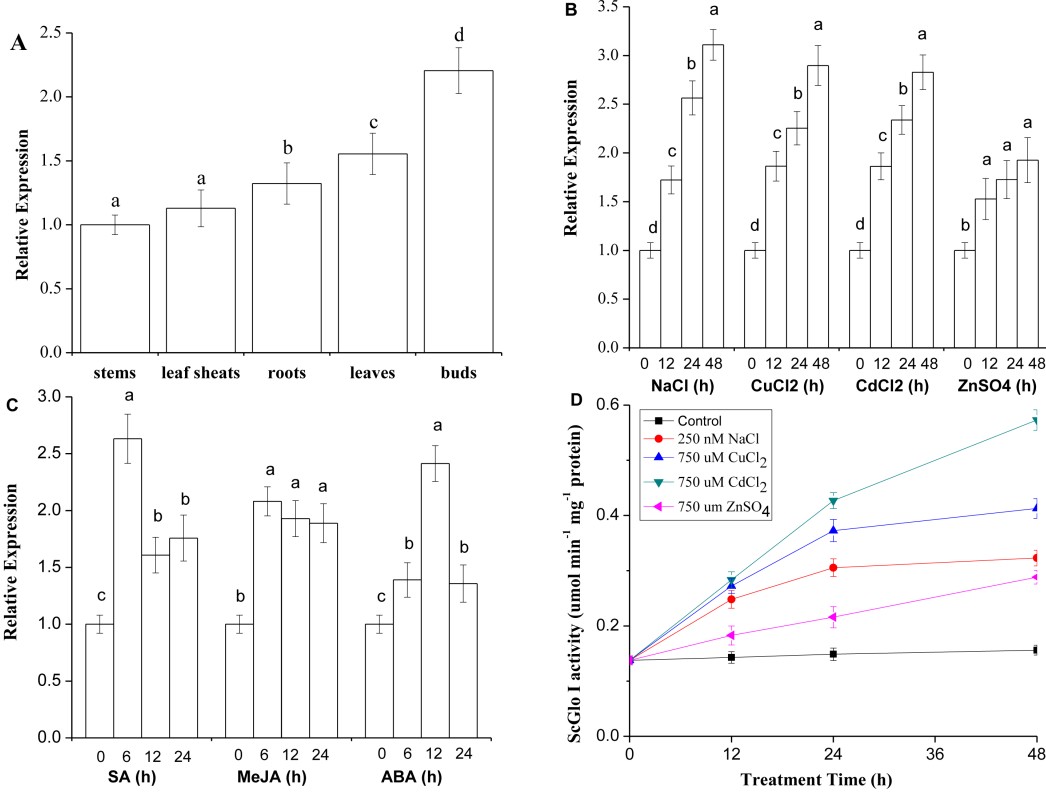

**Figure 5** **Expression patterns of *SoGloI* in sugarcane.** (A) *SoGloI* expression in different sugarcane tissues collected in the field; (B) *SoGloI* expression in greenhouse-grown sugarcane plantlets under NaCl, CuCl₂, CdCl₂, and ZnSO₄ treatments; (C) *SoGloI* expression in greenhouse-grown sugarcane plantlets under SA, MeJA, and ABA treatments; (D) glyoxalase I activity under NaCl, CuCl₂, CdCl₂, and ZnSO₄ treatments. Different letters indicate a significant difference at 5% level ($p \leq 0.05$). Each value represents the average of three biological repeats $\pm$ SE ($n = 3$).

through leakages in the glycolysis and TCA cycle (*Umea et al., 1994*). GlyI, the first enzyme of the glyoxalase system, plays a critical role in controlling MG levels and cytotoxicity (*Wu et al., 2013a*). The *GloI* gene has been cloned and characterized from several plant species. However, the glyoxalase I gene was never cloned and characterized in sugarcane. In the present study, a length *GloI* gene, designated as *SoGloI*, was isolated from a smut-resistant sugarcane cultivar YCE 05-179.

The GloI enzyme requires $Ni^{2+}$ or/and $Zn^{2+}$ for its catalytic activity (*Sukdeo et al., 2004*). *Sukdeo & Honek (2007)* reported that *Pseudomonas aeruginosa*, a gamma proteobacteria, encodes both $Ni^{2+}$ and $Zn^{2+}$ forms of the enzyme; GloA1, GloA2 (both Ni binding), and GloA3 (Zn binding). *Jain et al. (2016)* also found three active GLYI enzymes (AtGLYI2, AtGLYI3 and AtGLYI6) belonging to different metal activation classes coexisting in *Arabidopsis thaliana*. AtGLYI2 was found to be $Zn^{2+}$-dependent, whereas AtGLYI3 and AtGLYI6 were $Ni^{2+}$-dependent. $Ni^{2+}$-dependent *GloI* is present as a two-domain protein in all eukaryotes. Among the early branching eukaryotes, the group of algae appears to be the first to encode this gene (*Kaur et al., 2013*). In this study, a sugarcane *SoGloI* gene was

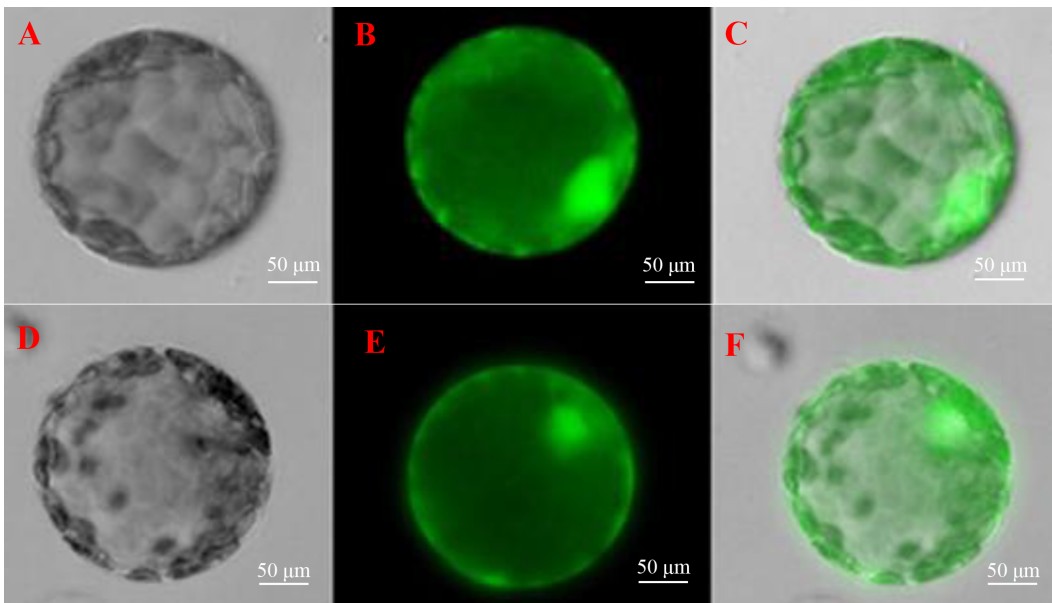

**Figure 6** **Determination of subcellular localization of ScGloI in tobacco (*Nicotiana benthamiana*) protoplasts under a fluorescence microscopy.** (A) bright field vision of pCXSN-GFP; (B) fluorescence vision of pCXSN-GFP; (C) merged vision of A and B; (D) bright field vision of pCXSN-SoGloI-GFP; (E) fluorescence vision of pCXSN-SoGloI-GFP; (F) merged vision of D and E. The emission wavelength is 515 nm and the excitation wavelength is between 470 nm and 490 nm.

found to encode two glyoxalase domains as well (Fig. S2). Besides, the multiple protein sequence alignment of SoGloI with those from other species indicated that SoGloI was a $Ni^{2+}$-dependent enzyme (Fig. 1). The result was similar to *OsGLYI*-11.2 (*Mustafiz et al., 2014*), who's expression was substrate inducible. However, unlike other eukaryotic $Zn^{2+}$-dependent glyoxalases, *OsGLYI*-11.2 is a $Ni^{2+}$-dependent monomeric enzyme.

Plant glyoxalase system in different tissues plays an important role at various vegetative and reproductive stages (*Mustafiz et al., 2011*). *GLOI* gene is required for cell division and proliferation; a higher enzyme activity has been found in rapidly dividing cells of cell suspensions, seedlings, and root tips (*Lin et al., 2010*; *Wu et al., 2013a*). In this study, *SoGloI* was constitutively expressed in various tissues of sugarcane genotype YCE 05-179, with the highest level in buds, followed by leaves, roots, leaf sheaths, and internodes (Fig. 5A).

To date, only a few reports have shown that *GLOI* gene is associated with disease resistance in plants. For instance, a maize *Glx-I* gene enhances the host defense against *Aspergillus flavus* through the detoxification of MG, a major product of *A. flavus* (*Chen et al., 2004*). The expression of wheat *TaGly I is* up-regulated 2.3-fold upon infection by *Fusarium graminearum* (*Lin et al., 2010*). In our previous study, *SoGloI* expression was up-regulated during infection with *S. scitamineum*, the pathogen of sugarcane smut (*Wu et al., 2013b*). In the present study, we used SA and MeJA to simulate biotic stress. Consistently, under SA and MeJA treatments, the *SoGloI* expression peaked at 6 hpt, when its activity reached 2.6- and 2.1-fold higher than that of the control, respectively (Fig. 5C). These results suggest that *SoGloI* expression can increase significantly under pathogenic

stresses; however, the exact role of *SoGloI* in pathogenic resistance process needs to be further investigated.

Glyoxalase I genes also have been implicated to enhance plant tolerance to salt stress. The expression of *Gly,* a glyoxalase I gene of *B. juncea,* is up-regulated after exposure to a high concentration of salt (*Veena, Reddy & Sopory, 1999*). The mRNA and polypeptide levels of *GLX1*, a glyoxalase I gene of tomato, increased by two to three folds in roots, internodes and leaves when the plants were treated with 10 g/L NaCl (*Espartero, Sanchez-Aguayo & Pardo, 1995*). The expression of two other glyoxalase I genes, B*v* M14-*glyoxalase I* of sugar beet (*Wu et al., 2013a*) and *TaGly I* of wheat (*Lin et al., 2010*), also significantly enhanced hosts' tolerance to salt stress. In this study, *SoGloI*-expressing Rosetta cells grown on agar plates tolerated high concentrations of NaCl up to 250 mM (Fig. 3B) and grew faster in LB liquid medium containing 250 mM NaCl (Fig. 4A). *SoGloI* expression was increased steadily from 0 to 48 hpt in sugarcane under salt stress (Fig. 5B). Under salt stress, glyoxalase I activity also elevated (Fig. 5D). Taken together, the results indicate the expression level of *SoGloI* can be significantly up-regulated under salt stress; however, more research is needed to reveal the underlying mechanism.

Zinc ($Zn^{2+}$), a micronutrient, is necessary for plant growth, but an excessive amount of $Zn^{2+}$ can inhibit plant growth (*Sun et al., 2006*; *Zarcinas et al., 2004*). A few studies have demonstrated that plant *GloI* genes enhance host tolerance to $Zn^{2+}$. Singla-Pareek et al. (2006) showed that *GlyI* from *B. juncea* enhanced host $Zn^{2+}$ tolerance to toxic levels in the transgenic tobacco. The expression of *TaGly I*, a glyoxalase I gene of *T. aestivum*, is induced continuously under 20 mM $ZnCl_2$ treatment. Compared to control, the increase in *TaGly I* expression is nearly 1.5-fold at 24 h (*Lin et al., 2010*). In the present study, *SoGloI*-expressing *E. coli* Rosetta cells were able to tolerate high concentrations of $ZnSO_4$ up to 750 µM (Fig. 3E) and also grew faster in LB liquid medium containing 750 µM $ZnSO_4$ (Fig. 4D).

Consistently, under ZnSO4 stress, the *SoGloI* expression in sugarcane was up-regulated steadily from 0 to 48 hpt, when its level and enzyme activity were 1.9-fold and 3.1-fold higher than that of the control (Figs. 5B, 5D). These results showed that *SoGloI* gene can enhance tolerance to excessive zinc stress even in a heterologous host system. Over-expression of glyoxalase I has been shown to confer tolerance to other heavy metals, such as cadmium or lead (*Singla-Pareek et al., 2006*). The level of expression and activity of *SoGloI* in *E. coli* (Figs. 3D, 4C) under $CdCl_2$ treatment (Figs. 5B, 5D) also supported this notion about tolerance to cadmium. Our work further showed that *SoGloI* expression and its enzyme activity were increased significantly under $CuCl_2$ treatment (Figs. 5B, 5D). All these findings suggest that *SoGloI* may be a good candidate gene for engineering to develop heavy metal resistant sugarcane cultivars.

As is known, sugarcane is a polyploidy and aneuploidy crop (*Scortecci et al., 2012*), in which low transformation efficiency remains one of the major limiting factors on transgenic sugarcane production (*Dal-Bianco et al., 2012*; *Gómez-Merino, Trejo-Téllez & Sentíes-Herrera, 2014*; *Scortecci et al., 2012*). This has also limited the functional analysis of isolated sugarcane genes; nonetheless, a model plant species (*Arabidopsis thaliana*, *Nicotiana benthamiana* or *Brachypodium distachyon*) with a shorter life cycle and simpler

genome can be explored as an alternative host for transforming and assessing the functional properties of isolated sugarcane genes, such as *SoGloI*.

## CONCLUSIONS

This is the first report on the cloning and characterization of glyoxalase I (*SoGloI*) gene in sugarcane. We isolated and characterized *SoGloI* gene and demonstrated the enzyme activity of glyoxalase I protein. We found that *SoGloI* expression and SoGloI enzymatic activity were elevated significantly when sugarcane tissues were subject to simulated biotic and abiotic stress conditions, such as high concentrations of salt or heavy metal ions. The findings have opened up a new research avenue for sugarcane to grow in polluted or salty environments via genetic engineering and breeding of *SoGloI* to enhance host resistance.

### Funding

This work was supported by the Natural Science Foundation of Fujian province, China (2015J06006), National Natural Science Foundation of China (31571732 and 31101196) and the Earmarked Fund for the Modern Agriculture Technology of China (CARS-20). The funders had no role in study design, data collection and analysis, decision to publish, or preparation of the manuscript.

### Grant Disclosures

The following grant information was disclosed by the authors:
Natural Science Foundation of Fujian province, China: 2015J06006.
National Natural Science Foundation of China: 31571732, 31101196.
Earmarked Fund for the Modern Agriculture Technology of China: CARS-20.

### Competing Interests

The authors declare there are no competing interests.

### Author Contributions

- Qibin Wu conceived and designed the experiments, performed the experiments, analyzed the data, prepared figures and/or tables, authored or reviewed drafts of the paper, approved the final draft.
- Shiwu Gao, Yachun Su and Jinlong Guo performed the experiments.
- Yong-Bao Pan authored or reviewed drafts of the paper, approved the final draft.
- Michael P. Grisham authored or reviewed drafts of the paper.
- Liping Xu and Youxiong Que conceived and designed the experiments, analyzed the data, contributed reagents/materials/analysis tools, authored or reviewed drafts of the paper, approved the final draft.

### Data Availability

The raw data has been supplied as a Supplemental File.

## Supplemental Information

Supplemental information for this article can be found online at http://dx.doi.org/10.7717/peerj.5873#supplemental-information.

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
