# Peer review of "Heterologous expression of a Glyoxalase I gene from sugarcane confers tolerance to several environmental stresses in bacteria"

_PeerJ, doi:10.7717/peerj.5873_

## Round 0.1 · original submission · Major Revisions

The referees have listed important points how the manuscript can be improved.

I have also checked the manuscript and suggest to carefully follow all suggested changes and additions before it will be acceptable. In particular, the following changes are required: Consider a new title as suggested. Rewrite the Results section in a structured way and in much more detail, including the appropriate reference to the Figures in order clearly and concisely present the obtained data. Also the Discussion should be extended and the recent references need to be added and also integrated. What is known from the other organisms should be included in more detail. As you generated recombinant enzyme, you might as well check its activity. In general, please check the language and check for typing errors. E.g.: line 41 six fold, line 64: B. juncea, line 73: what is ram alcohol?, line 79: salt-susceptible, line 246: 18.3-kDa His-tag, line 252: metal ion concentrations of up to, line 310: What means hpt?, line 352: Setaria italica, line 362: arid environments.

Reviewer 1 ·

Basic reporting

Literature reporting was not sufficient.

Experimental design

fine.

Validity of the findings

It could be further validated and improved.

Additional comments

The manuscript titled ‘A novel glyoxalase I gene from sugarcane plays an important role in response to environmental stresses’ by Wu et al provided detailed characterization of one sugarcane glyoxalase I member. The manuscript needs further modification to publish.
1. Firstly, the review was very back dated. The authors did not add any reference after 2013. While after 2013, there are a lot of other reports on glyoxalase pathway from rice, Arabidopsis, soybean and medicago plant. So try to incorporate the recent reports on glyoxalase pathway.
2. The result and discussion are very short. The authors could elaborate their findings and could compare with available information from other organisms.
3. The figures are not organized and seem to be scattered. Please concise the figures and try to reduce figure number.
4. Moreover, the authors have purified the recombinant protein in hand, but they not perform enzyme kinetics for the enzyme. They should perform enzyme kinetics of ScGlo1 and compare with previously reported members.

Reviewer 2 ·

Basic reporting

The manuscript reports characterization of a glyoxalase I enzymes from Sugarcane with potential role in stress response.
At some places, text is difficult to understand. For instance, lines 154, 162-163, 186, 196-198 and 234. Authors should correct grammatical errors and language.

In abstract as well as in introduction (line 94) and discussion (line 309), authors state that SA, MeJA, PEG and ABA treatments were given to mimic simulated pathogen stress conditions. This seems to be incorrect as PEG treatment is given to induce osmotic stress. And these hormones are not only involved in biotic stress but are important for abiotic stress response as well. Authors should be careful with their reporting and are advised to revise the text.
Importantly, authors refer to glyoxalase I from Sugar cane as ScGLo1 but the same name has been used for glyoxalases from S. cerevisiae also. Pl revise the nomenclature to avoid confusions.
Title should be modified to clearly indicate that response to stress has been tested in bacterial systems and not plants. In fact, title can be revised to “Heterologous expression of a Glyoxalase I gene from Sugarcane confers tolerance to salinity and heavy metal stresses in bacteria”.
Line 41: References should be cited in chronological order or cited just after the relevant context. Philips and Thornalley 1993 can be cited after the phrase “MG is a cytotoxic by-product produced mainly from triose-phosphate step’.
Line 45-48: The statement is incorrect. Glyoxalase system does not produces more free GSH, it uses and then recycles one molecule per cycle.
Lines 49-54: Authors have cited glyoxalases characterized from various species but have not mentioned recently characterized literature such as by Kaur et al. 2016, Plant J (rice), Mustafiz et al. 2014 Plant J (rice) Jain et al. 2016, Plos one (Arabidopsis).

Experimental design

Methods
Line 133: What does (+1) represents? Pl elaborate
Line 155: Why have ethanol and Tween 20 been used with 100 uM MeJA and 5 mM SA treatment?
Line 165: Why references have been cited after Table 1? Is it that sequences of primers have been retrieved from these references? If so, then include as footnotes in Table 1 otherwise remove these references.
Line 166: G-QF/G-QR represent primers of which genes?
Line 172: Correct to 2-ddct.
Line 209-210. Cite reference of transformation protocol.

Results
Line 230: Change the title to Sequence analysis of ScGLO1 (Note: revise the name of ScGlo1)
Figure 4: Why have authors shown growth curve in response to only NaCl and ZnSO4 and not CuCl2 and CdCl2. All should be shown. Y axis should be changed to “Absorbance (600 nm)” as it is not a direct measure of cell growth.
Figure 6: Authors have shown GLYI expression in response to different elicitors in two panels. PEG treated samples should be shown along with NaCl, CuCl2 and CdCl2 and not with SA, MeJA or ABA. The lower panel can be said to represent expression in response to hormonal treatments and upper panel representing abiotic stress treatments. Again, here now why have authors not shown expression upon ZnSO4 treatment. Same treatments should have been used in all experiments.
Line 271-276: Again, why have activities been demonstrated only in response to CdCl2 and NaCl and not ZnSO4 and CuCl2?
Also, what has been taken as 100% here, i.e. in what respect have the activities been calculated. What is the reference here?
Importantly, how can authors say that the activity they are measuring is specific to ScGlo1. Plants contain multiple glyoxalases. And thus, sugarcane is also possessing multiple glyoxalases. While measuring activity, we cannot distinguish between activities of specific glyoxalases. Authors should, thus, revise the section and state that they measured ‘specific Glyoxalase I activity’ in plant extracts.
Figure 8: Revise the caption to “Determination of Subcellular localization of ScGloI”. Details can follow after the caption.

Validity of the findings

Line 292: How can the presence of two glyoxalase domains confirm that it is an active glyoxalase I enzyme. The word “genuine” is incorrect and should be avoided. To determine whether it is a functional glyoxalase, authors should have checked the activity of purified glyoxalase I enzyme which will confirm that the gene which they transformed in E. coli is functional in nature and that tolerance is due to over-production of the corresponding protein. This will also determine whether this ScGlo1 is Ni or Zn dependent enzyme.
Lines 308-316: Mere upregulation of transcript levels does not mean that the gene has role in relieving stress. Authors should rephrase these statements.
Since, authors have characterized a two-domain glyoxalase I enzyme, they should cite reference of its recently characterized rice homolog OsGLYI-11.2 (Mustafiz et al. 2014, Plant J) indicating similarity between the two.
Line 329: Rephrase the sentence. Can’t conclusively state from the results that ScGlo1 plays an important role in sugarcane's response to salt stress as no transgenics have been raised and validated under salt stress here.

---

## Round 0.2 · Minor Revisions

The manuscript has been much improved, but there are still mistakes that have not yet been corrected. These errors and quite a few language problems are still not taken care of. In addition, some of the concerns from the previous reviews were not addressed in your revision. Therefore, another round of revisions is required, and all the changes should be listed, as well as answers to the open questions should be given in a separate letter to the editor.

Reviewer 2 ·

Basic reporting

Text still has many language mistakes. Authors should get the manuscript checked by a native English speaker.
References have been now cited but not discussed properly. Moreover, if more than one reference has been cited after a statement, these have not been arranged in the chronological order.

Experimental design

Discussed in the next section

Validity of the findings

Regarding response to previous queries:
Query: Figure 8: Revise the caption to “Determination of Subcellular localization of ScGloI”. Details can follow after the caption
Authors haven’t yet revised the figure caption.
Query: While measuring activity, we cannot distinguish between activities of specific glyoxalases. Authors should, thus, revise the section and state that they measured ‘specific Glyoxalase I activity’ in plant extracts.

Authors haven’t understood my query properly. What I meant was that the GLYI activity being tested here is contributed by all glyoxalase I enzymes of sugarcane (as sugarcane like other plants is having multiple glyoxalase I enzymes), how can authors say that they have specifically measured SoGLYI activity. This GLYI activity is contributed by other glyoxalase I enzymes as well which are present in sugarcane. So authors should state the activity as Specific GLYI activity (not referring to any particular enzyme i.e. SoGLYI in this case).

New queries:
Figure 5A: Describes expression in different tissues and not different stresses. Pl correct in figure legend.
Figure 5A, B, C: Authors have calculated relative expression in respect to which sample as control. What is the reference here? Pl specify.
Figure 5D: change x axis from treat time to treatment time. Secondly, “0” of x and y axis should coincide. Also, authors have not shown empty vector control’s activity in response to different heavy metal treatments which must be shown as it would serve as a control for comparison. Authors should include details about whether stressors were added right from the time zero i.e. start of the experiment or later.
Figure 6: Authors should have used DAPI dye as a nuclear marker as authors state that SoGLYI is getting localized to both nucleus and cytosol.
Lines 257-263: Since information about accession IDs of glyoxalases has been mentioned in figure 1 legend, authors need not repeat this information here. Can be omitted.
Line 275:18.3 kDa is not merely His tag but instead Trx-His-S-tag. Authors should correct it.
Line 299: Increased expression under stresses doesn’t mean that SoGloI confers stress tolerance. Authors should revise or omit this sentence.
Line 304: Authors have mentioned heavy metal treatments as stresses everywhere in the text, as also written in this particular sentence. Authors should refer these as treatments and not stresses everywhere in the text. Though these treatments confer stress on plants but there are no terms as CuCl2 stress etc.
Discussion: many repetitions are there in discussion section like in Lines 324-333 and 334-340. These have already been mentioned in the introduction section. Pl revise. Also carefully, check the entire section for other possible repetitions.

Line 365: ABA mainly mimics abiotic stress, how can then authors say that it stimulates specifically pathogenic stresses. Pl revise.
Line 367: Again, ABA is a treatment and not stress. Pl revise
Line 387-389: What do authors mean from “hosts’ tolerance to Zn and Zn accumulation”. How is Zn and Zn accumulation separate? Pl rephrase.
Line 407-408: Statement is incorrect. Glyoxalase system doesn’t recycle GSH with MG to form hemithioacetal. Instead, GSH spontaneously forms a hemithioacetal adduct with MG, which is the actual substrate for glyoxalase I enzyme. GSH is used in the first reaction catalyzed by glyoxalase I and recycled back in the second step catalyzed by glyoxalase II.
Figure 2: Change the caption to “Heterologous expression of recombinant SoGloI in E. coli cells”. Following the caption, details can be mentioned.
Figure 4: Change caption to “Growth assessment of SoGloI overexpressing E. coli cells to salt and heavy metal treatments. Following which, details should be mentioned about the time stressors were added and their concentrations.

---

## Round 0.3 · accepted · Accept

The manuscript has now been sufficiently improved.

One error.
line 324: please correct the phrase to either "algae appear......" or "the group of algae.....".

#